# Benchmarking Out-of-Distribution Generalization Capabilities of DNN-based Encoding Models for the Ventral Visual Cortex.

**Spandan Madan**
Harvard University

**Will Xiao**
Harvard Medical School

**Mingran Cao**
Francis Crick Institute

**Hanspeter Pfister**
Harvard University

**Margaret Livingstone**
Harvard Medical School

**Gabriel Kreiman**
Harvard Medical School

## Abstract

We characterized the generalization capabilities of deep neural network encoding models when predicting neuronal responses from the visual cortex to flashed images. We collected *MacaqueITBench*, a large-scale dataset of neuronal population responses from the macaque inferior temporal (IT) cortex to over $300,000$ images, comprising $8,233$ unique natural images presented to seven monkeys over $109$ sessions. Using *MacaqueITBench*, we investigated the impact of distribution shifts on models predicting neuronal activity by dividing the images into Out-Of-Distribution (OOD) train and test splits. The OOD splits included variations in image contrast, hue, intensity, temperature, and saturation. Compared to the performance on in-distribution test images—the conventional way in which these models have been evaluated—models performed worse at predicting neuronal responses to out-of-distribution images, retaining as little as $20\%$ of the performance on in-distribution test images. Additionally, the relative ranking of different models in terms of their ability to predict neuronal responses changed drastically across OOD shifts. The generalization performance under OOD shifts can be well accounted by a simple image similarity metric—the cosine distance between image representations extracted from a pre-trained object recognition model is a strong predictor of neuronal predictivity under different distribution shifts. The dataset of images, neuronal firing rate recordings, and computational benchmarks are hosted publicly at: MacaqueITBench Link.

## 1 Introduction

Deep Neural Networks (DNNs) for vision have internal representations that purportedly share similarities with neural representations in the primate ventral visual cortex stream [2, 3]. Such correlations between the representations in artificial and biological neural networks allow for models that use image representations extracted from a pre-trained DNN (e.g., ResNet [4]) to predict neuronal firing rates [5] (Fig. 1(a)). However, DNNs are known to struggle with generalization under distribution shifts such as Out-of-Distribution (OOD) viewpoints [6, 7, 8], materials and lighting [9, 10], and noise [11, 12]. The problem of OOD generalization constitutes a key standing challenge in computer vision. Here we investigate whether this difficulty in generalization also affects models of the visual cortex that rely on a DNN to extract image representations.

We hypothesize that, even within an image set where DNN-based models predict neural responses well under random splits across images, specific train-test splits with distribution shifts will impair model performance, proportional to the size of distribution shift. To test this hypothesis, we collected

38th Conference on Neural Information Processing Systems (NeurIPS 2024) Track on Datasets and Benchmarks.

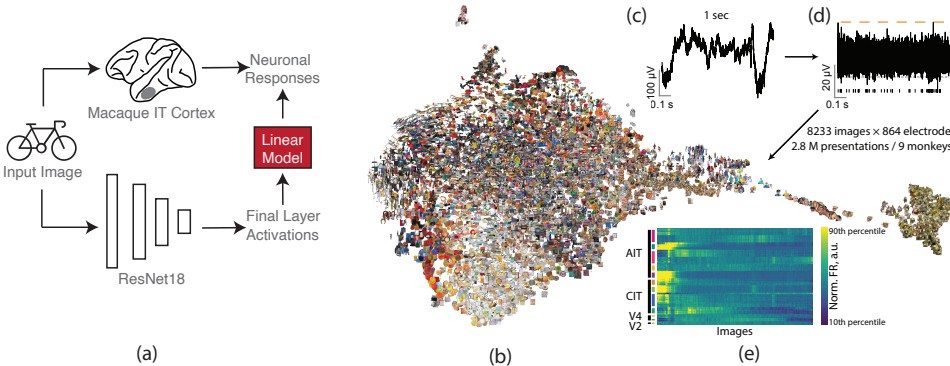

Figure 1: *Modeling the ventral visual cortex with MacaqueITBench.* (a) DNN-Based models of the visual cortex employ a linear model to map image features extracted from pre-trained DNNs (e.g., ResNet18) to neuronal responses collected from the macaque cortex (e.g., IT cortex). (b) UMAP [1] visualization of the representation of images by the neuronal pseudo-population. Nearby images have more similar population responses. (c) An example one-second segment of the raw wideband signals recorded on an electrode. (d), The wideband signals were highpass filtered, and threshold-crossing events below a voltage value (dashed black line) were counted as multiunit spikes (lower vertical ticks). The orange horizontal bars indicate image presentation periods. (e) The heatmap shows the neuronal response matrix. Each row indicates the responses from an electrode, pooled across sessions. The columns correspond to images, sorted by the reverse UMAP horizontal order. The vertical bars to the left of the heatmap denote the recorded areas (black lines) and monkeys (colored lines).

*MacaqueITBench*, a large-scale dataset of responses to natural images by neurons in the macaque ventral visual pathway. The dataset comprises neurons in V2, V4, Central IT (CIT), and Anterior IT (AIT) (primarily CIT and AIT) and includes responses to over $300,000$ images ($8,233$ unique images presented to seven monkeys over 109 sessions), as illustrated in Fig. 1(b).

Using *MacaqueITBench*, we investigated the impact of distribution shifts on the neural predictivity of DNN-based models of the visual cortex. We systematically constructed various OOD distribution shifts, some of which are schematized in Fig. 2. Foreshadowing, our main finding is that distribution shifts in even low-level image attributes break DNN-based models of the visual cortex.Furthermore, the relative ranking of different models, usually considered as a key metric to compare models, is *not* conserved across distribution shifts. These observations highlight a fundamental problem in modern models of the ventral visual cortex—good predictions are limited to images similar to those in the training data distribution.

To explain the OOD model-performance drop, we built on theoretical work positing that generalization performance is closely correlated with the amount of distribution shift [13, 14]. While theoretical studies have examined simplistic, simulated data, we show that a suitable metric of the size of distribution shifts can account for the OOD generalization performance of neural-encoding models.

In summary, our main contributions are:

- We present *MacaqueITBench*, a large-scale dataset of neural population responses to over $300,000$ images spanning multiple areas of the primate ventral visual pathway. The recording included 640 electrodes (12 multi-electrode arrays) recorded in nine hemispheres of seven monkeys.

- We show that modern models of the visual cortex do not generalize well—simple distribution shifts can reduce neural predictivity to as low as $20\%$ of in-distribution performance.

- We show that the ranking across models is not conserved across distribution shifts.

- We provide a simple metric of distribution shift size that captures neural predictivity changes under distribution shifts.

## 2 Related Work

### 2.1 DNN-based models of the ventral visual cortex

A touchstone for visual neuroscience is the ability to predict neuronal responses to *arbitrary* images. On this test, DNN-based models have emerged as state-of-the-art models, best explaining neuronal responses across species—mouse, macaque, and humans—and visual cortical areas—from the primary visual cortex (V1) to the high-level inferior temporal cortex (IT) (for review, see [15, 16, 17]). These DNN-based models have been evaluated using random cross-validation (e.g., [18]), which tests IID generalization typically within a rather homogeneous image set. OOD generalization in such models has been sparsely examined. One study compared model fit to neural responses on two image types [19]. Here we systematically vary the type and degree of OOD splits to assess generalization as a function of differences between training and test datasets.

### 2.2 Out-of-distribution generalization capabilities of DNNs

In computer vision, DNNs for object recognition have been documented to fail at generalizing across a wide range of distribution shifts. Such shifts include 2D rotations and shifts [20, 21], commonly occurring blur or noise patterns [11, 22, 23, 24], and real-world changes in scene lighting [25, 26, 27], viewpoints [7, 28, 29, 30, 25, 8, 31], geometric modifications [32, 33, 34], color changes [35, 36], and scene context [37, 38].

Several benchmarks have been proposed to capture these distribution shifts systematically. For handwritten digit recognition, datasets like MNIST [39], MNIST-M [40], SVHN [41], and SYN [40] differ in features such as font, color, and background. For object recognition, domain shifts in the form image style have been captured in datasets like VLCS [42], Office-31 [43], and PACS [44]. Similarly, the Terra-Incognita [26] dataset has captured domain shift between the same scene viewed under daylight and night conditions. Recently, the WILDS benchmark [45] was introduced to tackle distribution shifts encountered in real-world scenarios, featuring datasets in diverse fields like animal and molecule classification. Of note, there has also been some work using controlled synthetic data to generate systematic benchmarks for generalization. These include the Biased-Cars dataset [7], the human visual diet dataset [9], and the Photorealistic Unreal Graphics (PUG) datasets created using Unreal Engine [46].

There have been three broad approaches to address the lack of OOD generalization in DNNs: first, modifying the learning paradigm including modifying the architecture or loss function to enforce invariant representations [47, 48, 49, 50, 51], or using ensemble and meta-learning [52, 53, 54]; second, modifying the training data using data augmentation [55, 56, 57, 58], or by increasing data diversity [23, 59, 60, 61, 62, 9, 7, 63, 6]; third, scaling data up to beyond billions of data points [64, 65, 66]. Despite these efforts, OOD generalization remains an unsolved problem in computer vision.

### 2.3 Out-of-distribution generalization models of the visual cortex

Despite extensive machine learning research on the topic, OOD generalization has received limited attention in the context of modeling biological neuronal responses. The ability to generalize is especially relevant to ventral visual cortex models due to acute limitations on the amount of available neuronal data. Given the time needed to present images (100s of ms per image), finite neuronal recording durations, and repeat presentations needed to combat neuronal stochasticity, it is currently infeasible to collect reliable neuronal responses to many more than 10k unique images. In this data-limited regime, most images of interest will remain out-of-domain even if we had foreknowledge of the test distribution (e.g., 10k unique images equal 10 images per category for the 1,000 ImageNet categories, insufficient to cover the distribution). The limited OOD generalization ability of current neuronal encoding models restricts their scientific utility, for example in accuratelypredicting maximally activating images for neurons (Fig. S1). This work contributes by borrowing from machine learning research on OOD generalization to shed light on computational neuroscience models.

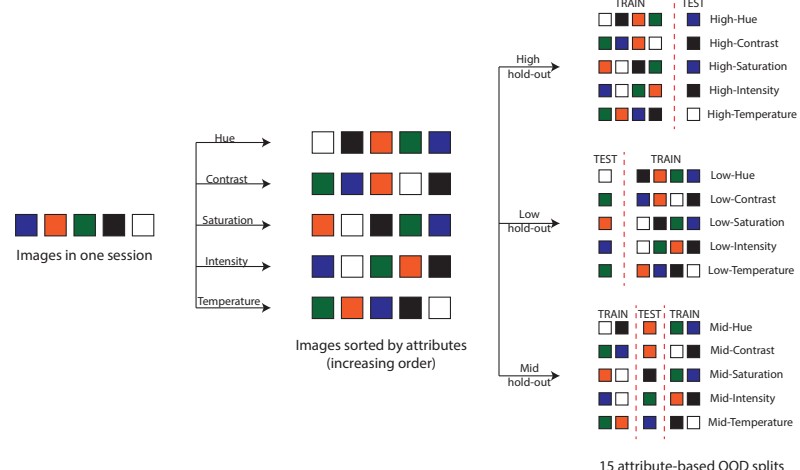

Figure 2: *Constructing multiple attribute-based OOD splits.* For each of our 109 sessions, we constructed 15 different attribute-based OOD splits. These splits correspond to 3 hold-out strategies (*high, low, mid*) for each of 5 image-computable attributes (hue, contrast, saturation, intensity, temperature). For each attribute (e.g., hue), we compute the attribute value for each image in the session. For the *high* hold-out strategy, all images with the attribute value above a percentile cut-off serve as the OOD test set with the remaining serving as the train set. Analogously for the *low* hold-out splits, images below a percentile cut-off serve as the test set with the remaining serving as the train set. For *mid* hold-out splits, images within the middle percentiles serve as the test set.

## 3 MacaqueITBench: Image-response recordings from the ventral stream

We collected a large-scale dataset of neuronal population responses to over $300,000$ images across sessions, comprising $8,233$ unique natural images presented to seven monkeys over 109 sessions. In each session, a monkey maintained fixation while images were rapidly presented in random order. Each presentation was 83 milliseconds; with 83–150 milliseconds between presentations.

The images were derived from published image sets [67] and photos taken in the lab and contained pictures of common objects, people, and other animals including monkeys (Fig. 1(b)). Image thumbnails are shown in Fig. 1(b)); sample images are provided in Fig. S2. Images belonged to over 300 semantic categories annotated by hand. A full list of categories can be found in Table S1. The large number and diversity of images allowed us to construct various OOD splits.

## 4 Constructing out-of-distribution data splits

We build on past work studying generalization under systematic distribution shifts [7, 9, 47, 11], and define the training and test distributions parametrically using image attributes. Using these parametric data distributions, we construct three kinds of train-test splits:

**InDistribution (InD) splits**: For each session, we created one In-Distribution (InD) split to compare with OOD generalization performance. We sampled $25\%$ of the images at random, and held these out as the InD test set, with the remaining images serving as the training set.

**Attribute-based OOD splits:** We describe here OOD splits based on image contrast; splits based on the other image attributes were constructed analogously. For each session, we computed the contrast value for each image. Then, one of three strategies were employed (Fig. 2):

- *High hold-out*: The $75^{th}$ percentile of contrast values served as the cut-off. Images with contrast above the cut-off formed the test set. Remaining images formed the training set.

- *Low hold-out*: The $25^{th}$ percentile served as the cut-off. All images below this cut-off served as the held-out test set. The remaining images served as the training set.

- *Mid hold-out*: Images with contrast values between the $37.5^{th}$ and $62.5^{th}$ percentile served as the held-out test set. The remaining images formed the training set.

**Cosine Distance-based splits:** To investigate the relationship between the size of distribution shift and neuronal response predictivity, we constructed 3 additional test splits. We first extracted the features for every image from the pre-final layer of a pre-trained ResNet18. A random image was picked to be the seed, and all images in the session were sorted in order of increasing cosine distance between the ResNet extracted features of the images and the seed. The sorted images were then divided into three chunks based on percentile cut-offs. The first chunk corresponded to the bottom $80^{th}$ percentile which served as the Training + In-Distribution Test split. A random subset of this first chunk was held out to form the In-Distribution test split, with the remaining serving as the training set. The second chunk included images in the $90^{th}$ to $95^{th}$ percentile, which were held-out as the *Near-OOD* test split. Finally, the third chunk corresponded to images above the $95^{th}$ percentile. These were held-out as the *Far-OOD* split. To ensure a gap between the train and test distributions, we did not consider images between the $80^{th}$ and the $90^{th}$ percentile. Note that the number of images in the In-Distribution test split was kept the same number of images as the Near-OOD split.

# 5 Quantifying distribution shifts

We present a unified framework for measuring distribution shifts over the parametric OOD train-test splits presented in Sec. 4.

## 5.1 Representations for training and testing data-splits

Let $D_T = \{i_1^T, i_2^T, ..., i_N^T\}$ denote a train split of $N$ images, and let $D_t = \{i_1^t, i_2^t, ..., i_n^t\}$ denote the corresponding test split of $n$ images. $\mathcal{R}(.)$ is a representation function that provides a vector representation for an image. The train and test images thus correspond to $\mathcal{R}(D_T) = \{\mathcal{R}(i_1^T), \mathcal{R}(i_2^T), \ldots, \mathcal{R}(i_N^T)\}$ and $\mathcal{R}(D_t) = \{\mathcal{R}(i_1^t), \mathcal{R}(i_2^t), \ldots, \mathcal{R}(i_n^t)\}$, respectively.

We analyzed representations $\mathcal{R}(i_j)$ formed by the features extracted for an image $i_j$ by a pre-trained DNN. We evaluated 8 different DNN architectures, and multiple layers for every architecture. The equations below are agnostic to the architecture and the layer used. Other alternatives could include using HOG [68] or GIST [69] image features, or the vectorized pixel values of the image.

## 5.2 Defining distances over different datasets

To compute the shift between $\mathcal{R}(D_T)$ and $\mathcal{R}(D_t)$, we compared three distance metrics:

**Maximum Mean Discrepancy ($D_{\textbf{MMD}}$):** The MMD distance between the two datasets can be computed as

$$D_{\text{MMD}}^2(D_T, D_t) = \frac{1}{N^2} \sum_{j=1}^{N} \sum_{k=1}^{N} K(\mathcal{R}(i_j^T), \mathcal{R}(i_k^T)) + \frac{1}{n^2} \sum_{j=1}^{n} \sum_{k=1}^{n} K(\mathcal{R}(i_j^t), \mathcal{R}(i_k^t))$$
$$- \frac{2}{Nn} \sum_{j=1}^{N} \sum_{k=1}^{n} K(\mathcal{R}(i_j^T), \mathcal{R}(i_k^t))$$

Here, $K(\mathcal{R}(i_j^T), \mathcal{R}(i_k^t))$ is a kernel distance between the representations of images $i_j^T$ and $i_k^t$. For our experiments, we used a Gaussian RBF kernel.

**Covariate-Shift ($D_{\textbf{Cov}}$):** Let $P_T(X)$ and $P_t(X)$ denote the distributions of the train and test input variables (i.e., image representations), and let $P(Y|X)$ denote the conditional distribution of the output (i.e., neuronal responses) given the input. A covariate shift exists if $P_T(X) \neq P_t(X)$ but $P_T(Y|X) = P_t(Y|X)$. $D_{\text{Cov}}$ can be computed by training a binary classifier to classify if data comes from the training or the testing dataset. We denote the accuracy of this classifier as $a_{T,t}$ and measure the covariate shift as:

$$D_{\text{Cov}}(D_T, D_t) = 2 \times (0.5 - a_{T,t}).$$

**Closest Cosine Distance** ($D_{\textbf{CCD}}$)**:** For every image in the test set, we find its distance to the closest training image, and compute the mean of this distance over all test images. For brevity, we will refer to this as *Closest Cosine Distance*. Let $i_k^T \in D_T$ denote the closest training image to test image $i_j^t \in D_t$ as measured by the cosine distance $D_{\cos}(\mathcal{R}(i_j^T), \mathcal{R}(i_k^t))$. The distance $D_{\cos}$ between two vectors $u$ and $v$ is given by:

$$D_{\cos}(u, v) = 1 - \frac{u \cdot v}{\|u\|\|v\|}$$

The average distance to the closest training image is:

$$D_{\text{CCD}} = \frac{1}{n} \sum_{j=1}^{n} \min_{k \in \{1,2,\ldots,N\}} D_{\cos}(\mathcal{R}(i_j^T), \mathcal{R}(i_k^t))$$

## 6 Model training and evaluation

As depicted in Fig. 1(a), we used a linear model to map pre-trained model activations to neuronal firing rates from the IT cortex (Fig. 1(a)). The linear model was learned using ridge regression. We used only pre-trained DNNs, not DNNs fine-tuned for our analysis.

For feature extraction, we investigated 8 DNN architectures and 2 layers for each architecture. The DNNs include supervised models trained on ImageNet (ResNet-18 [4], ViT [70]), self-supervised models trained on billion-scale data with self-supervised and weakly supervised learning (ResNet18_swsl [64], ResNext101_32x16d_swsl [64], ResNet-50_ssl [64]), Noisy student with EfficientNet [71], self-supervised learning over billions of tokens (DinoV2 [66]), and the multi-modal vision-language model CLIP [65]. The exact layer used for feature extraction for each model is provided in the supplement in Sec. D.

A linear encoding model was fit for the trial-averaged responses of each neuron in a session. The results are presented as the mean and S.E.M. across 109 sessions (7 monkeys); each session's results is the median across neurons. The model fit per neuron was quantified as the ceiling-normalized, squared Pearson's correlation, $r_{\text{pred}}^2 / r_{\text{cons}}^2$ following convention [18, 72] and related to the explained variance, $R^2$. The ceiling $r_{\text{cons}}$ of a neuron was calculated as its response correlation between split-half trials, across images, with Spearman-Brown correction (because model fitting used all trials per image). The model fit $r_{\text{pred}}$ was the correlation across test images between neuronal responses and model predictions. All experiments were conducted on a compute cluster with 300 nodes, 48 cores per node with CPU machines running Rocky Linux release 8.9 (Green Obsidian).

## 7 Results

### 7.1 Neural predicitivity drops under distribution shifts

DNN-based encoding models become worse at predicting neuronal responses under simple shifts in the image distribution. To demonstrate this, we report the ratio of neural predictivity between OOD and In-Distribution test splits ($r_{ood}^2 / r_{ind}^2$). A ratio of 1 would indicate that models generalize equally well to InD and OOD test images (horizontal dashed line; Fig. 3a). In contrast, the OOD/InD performance ratios are substantially lower than 1. For instance, the black bar in Fig. 3a shows that the model's neural predictivity was 0.33 on *high*-hue OOD images (constructed using the *high hold-out* strategy in Sec. 4) compared to images with InD hue. Models show a similar lack of OOD generalization to OOD images with regard to saturation (red bar), intensity (green bar), temperature (blue bar), and contrast (gray bar). This performance drop was observed for all eight DNNs tested (Fig. 3b-h) and ranged from a best-case ratio of 0.66 for the CLIP model generalizing to *high*-temperature OOD images to a worst-case ratio of 0.2 for the ViT model generalizing to *high*-saturation OOD images.

The lack of OOD generalization by neuron encoding models extended to models based on intermediate DNN layers, not just the penultimate layer. For all eight models, using activations extracted from

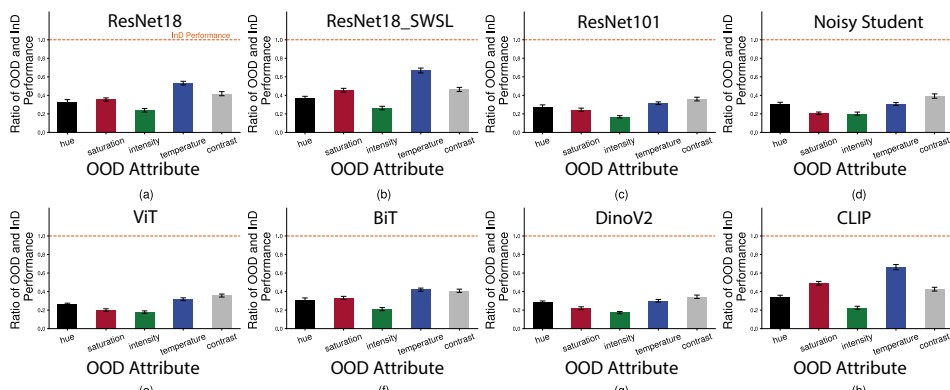

Figure 3: *Neuronal response predictivity drops under distribution shifts.* The y-axis shows the ratio of the neuronal response predictivity for out-of-distribution (OOD) images to in-distribution (InD) test images. A ratio of 1 would indicate no drop in performance. Each panel (a-h) shows a different architecture used for extracting image features. Each bar in a panels corresponds to a different OOD split constructed by using the *high* hold-out strategy across 5 different attributes (hue, saturation, saturation, intensity, temperature, and contrast). For all architectures and OOD splits, models fail to generalize well to OOD samples and are significantly and substantially below the 1.0 horizontal line. Image features were extracted from the pre-final layer for all architectures. Error bars denote standard deviation.

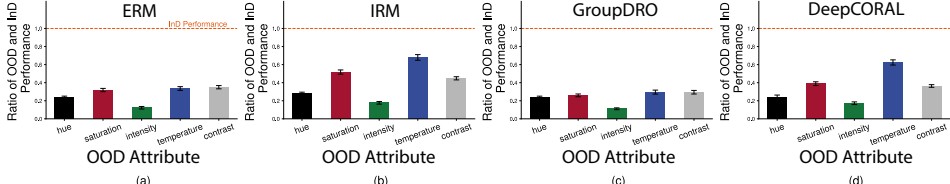

Figure 4: *Neuronal response predictivity drops for algorithms specifically designed to tackle OOD generalization as well.* Neuronal response predictivity is reported on OOD test splits constructed using the *high* hold-out strategy (using the same format as in Fig. 3). Generalization performance is well below 1.0 across image-computable attributes and four algorithms designed for OOD generalization presented in past literature [73, 45]. Specifically the four panels respectively show results for a ResNet50 model trained with empirical risk minimization (ERM) [74], Invariant Risk Minimization (IRM) [47], GroupDRO [75], and DeepCORAL [76] algorithms. None of these models generalizes well to OOD splits constructed with the *high* hold-out strategy despite being designed specifically for OOD generalization.

intermediate layers (layer names shown in Fig. 5), OOD performance remained substantially lower than InD performance (Fig. 5; tabular form in Sec. E).

Our findings extend to specialized Domain Generalization architectures designed to be more robust to distribution shifts (Fig. 4). For all specialized architectures and image attributes, the ratio of OOD and In-Distribution performance was significantly below 1.0, confirming a sharp drop in neural predictivity under distribution shifts.

OOD model performance was consistently lower than InD performance across hold-out strategies. Fig. 6 shows the OOD/InD model performance ratio for OOD splits constructed using the *low* hold-out strategy described in Sec. 4. As before, the ratio is consistently below 1.0, which confirms a severe drop in neural predictivity under distribution shifts. This finding also held true for Domain Generalization architectures tested with the low hold-out strategy, and for additional OOD shifts constructed using the mid hold-out strategy as shown in supplementary Sec. F.

So far, we have presented results with OOD shifts constructed by holding-out images within specific ranges of image attributes including hue, contrast, saturation, and color temperature. These splits model realistic scenarios where a model must generalize to, for example, novel weather and lighting

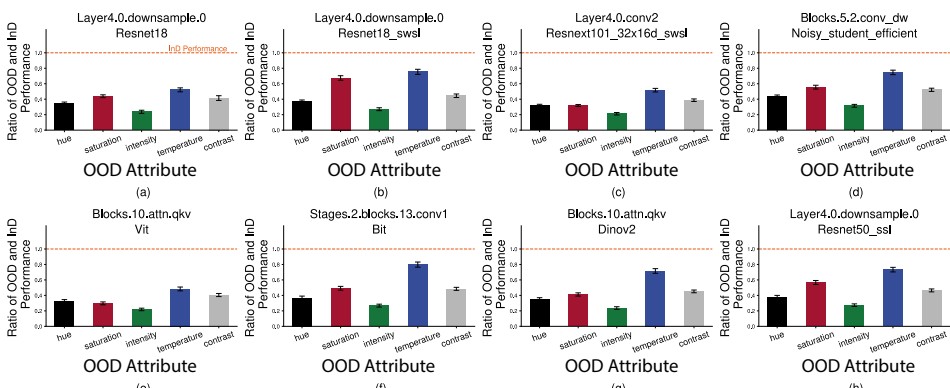

Figure 5: *Neuronal response predictivity drops under OOD testing for different model layers as well.* Neuronal response predictivity on OOD samples is reported for multiple DNN architectures across multiple different layers. Layer name is mentioned alongside architecture in all panels (a-h). All OOD splits reported here were constructed using the *high* hold-out strategy. For all architectures, layers, and OOD splits, models fail to generalize well to OOD samples and are significantly below the 1.0 horizontal line.

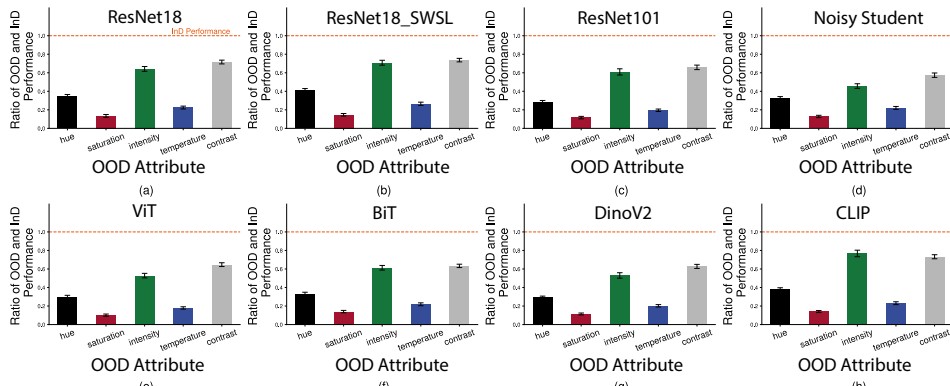

Figure 6: *Neuronal response predictivity drops for the low hold-out strategy as well.* Neuronal response predictivity is reported on OOD test splits constructed using the *low* hold-out strategy. Across all DNN architectures and image-computable attributes, performance is below 1.0 for all panels (a-h). Thus, models do not generalize well to OOD splits constructed with the *low* hold-out strategy either.

conditions. To validate our results in other scenarios of domain shift, we examined two additional splits based on categories. First, we held out a random subset of categories as a test set and trained on the remaining categories. Second, we held out Food-related categories as a test set and trained on non-Food categories. For both these OOD splits, all models failed to generalize—ratio of OOD and In-Distribution neural predictivity was well below 1.0, in line with previous results (Table 1).

Finally, we also compared neural predictivity on an established, in-distribution benchmark (Brain-Score) and on our OOD benchmark. These results, presented below, further support our finding that current DNNs are insufficient models of the Ventral Visual Cortex—models that perform better on the in-distribution BrainScore benchmark did not perform better on OOD shifts (all Spearman rank correlations p > 0.05). Combined, these results showcase a problem for current DNN-based models of the visual cortex—despite their ability to predict neural responses to in-distribution test images, the models generalize poorly under distribution shifts even in low-level image attributes.

| OOD Split | CLIP | DinoV2 | Noisy Student | ResNet18 | ResNet50 SSL | ResNext 101 |
|---|---|---|---|---|---|---|
| Random | $0.80 \pm 0.01$ | $0.77 \pm 0.02$ | $0.78 \pm 0.02$ | $0.83 \pm 0.01$ | $0.80 \pm 0.01$ | $0.71 \pm 0.01$ |
| Food | $0.23 \pm 0.02$ | $0.19 \pm 0.01$ | $0.20 \pm 0.02$ | $0.23 \pm 0.02$ | $0.20 \pm 0.02$ | $0.16 \pm 0.01$ |

Table 1: *Similar conclusions are reached with naturalistic OOD splits*. This table shows the ratio of neuronal response predictivity for OOD samples to in-distribution samples, which is below 1.0 for all architectures. The Random split was constructed by holding out a random subset of categories as the test set, and training the model on remaining categories. For the Food split, all food-related categories served as the tet set, and models were trained on the non food-related categories.

| Model | Brain Score | Hue | Saturation | Intensity | Temp | Contrast | Average |
|---|---|---|---|---|---|---|---|
| BiT | 0.33 | 0.31 | 0.33 | 0.21 | 0.42 | 0.41 | 0.33 |
| ResNet18 | 0.35 | 0.33 | 0.36 | **0.24** | 0.53 | 0.42 | 0.37 |
| CLIP | 0.47 | **0.34** | **0.49** | 0.23 | **0.66** | **0.43** | **0.43** |
| ResNext101 | 0.49 | 0.28 | 0.25 | 0.17 | 0.32 | 0.36 | 0.32 |
| ViT | **0.51** | 0.26 | 0.20 | 0.18 | 0.32 | 0.36 | 0.30 |

Table 2: *BrainScore vs MacaqueITBench*. We compare models of the visual cortex in and outside the training data distribution. BrainScore [18] provides a ranking of models based on in-distribution performance. However, models that perform better on the in-distribution BrainScore benchmark did not perform better on OOD shifts (all Spearman rank correlations p > 0.05). Best performing model has been presented in bold.

## 7.2 The distance between train and test distributions explains generalization performance

The results above raise a natural question—when and how do models of the ventral visual cortex fail to generalize under distribution shifts? Theoretical work has related OOD generalization to the amount of distribution shift [13, 14]. Here we apply this theoretical framework to characterize generalization in DNN models of the brain.

Intuitively, model generalization should be worse for train-test splits under larger distribution shifts. We tested this intuition by constructing splits with different levels of distribution shifts—InD, Near OOD, and Far OOD. As described in Sec. 4, images in every session were sorted based on cosine distance and split into three chunks. The first chunk formed the training set and the In-Distribution test set, while the second and third chunks formed the Near OOD and Far OOD test sets. As hypothesized, model performance decreased significantly from In-Distribution to Near OOD, then Far OOD test sets (Fig. 7(a); two-sided t-test, $p < 0.01$).

The size of the distribution shift predicted the OOD model performance drop across individual data splits (Fig. 7(b)). The distribution shift between each pair of train and OOD test distributions was quantified with the *Closest Cosine Distance* ($D_{CCD}$; described in Sec. 5). The $D_{CCD}$ strongly correlated with the OOD model performance drop (Spearman correlation $\rho = -0.49$).

The distribution shift ($D_{CCD}$) calculated from ResNet features also explained OOD performance for attribute-based splits (Fig. 7(c). Across all image attributes (hue, saturation, temperature, contrast, intensity) and hold-out strategies (*low, high, mid*) used to create OOD splits, $D_{CCD}$ correlated with OOD model performance drop (Spearman correlation $\rho = -0.45$). Compared to two other popular measures of the sizes of distribution shifts (MMD, $D_{MMD}$[77] and Covariate-Shift, $D_{Cov}$ [78]; Sec. 5), $D_{CCD}$ best predicted OOD model performance (Fig. 7(d)).

## 8 Conclusions

These results reveal a deep problem in modern models of the visual cortex: good prediction is limited to the training image distribution. Simple distribution shifts break DNN models of the visual cortex,

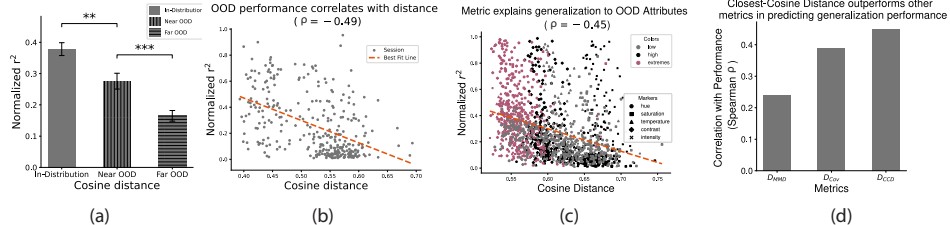

Figure 7: *Closest-Cosine Distance metric well explains performance across all attribute-based OOD splits.* (a) Neural predictivity on distance-based splits. Models performed best on the In-Distribution (InD) test split, dropping in performance from InD to Near OOD test set and from Near OOD to Far OOD (both $p < 0.01$, two-sided t-test). This suggests a relationship between the extent of distribution shift and generalization performance. (b) OOD performance can be well-explained by the distribution shift. For all 109 sessions, the plot shows performance on the InD, Near-OOD, and Far-OOD with the corresponding distribution shift measured using the Closest-Cosine Distance metric ($D_{\text{CCD}}$). Performance and $D_{\text{CCD}}$ have a Spearman correlation of $-0.49(p < 0.001)$. (c) Scatter plot of neural predictivity and the corresponding distribution shift ($D_{\text{CCD}}$) across all 15 attribute-based OOD splits for all 109 sessions. Generalization performance and the proposed distance metric have a Spearman correlation of $-0.45(p < 0.001)$ (d) Comparing different distance metrics w.r.t. their correlation with OOD performance. The proposed Closest-Cosine Distance has the highest correlation with neural predictivity, outperforming both MMD ($D_{\text{MMD}}$) and Covariate-Shift ($D_{\text{Cov}}$).

consistent with broader findings that the underlying DNNs are brittle to OOD shifts. Going one step further, we introduce an image-computable metric that significantly predicts the generalization performance of models under distribution shifts. This metric can help investigators gauge how well a neural model fit on one dataset may generalize to novel images.

Our findings underline an important limitation of AI models for Neuroscience. Fields like Computer Vision have responded to the issue of distribution shifts by collecting progressive larger datasets, hoping models will learn to generalize to most images [79, 80, 81, 82] at the billion-image scale. However, it is infeasible to achieve the same scale in neuroscience—the time needed to present a billion images is already a formidable challenge, not to mention the resource intensiveness of data collection. We hope our characterization of when and how modern models of the visual cortex fail out-of-domain will motivate the development of data-efficient ways to improve DNN generalization.

## 9   Limitations

In this work, we have explored the impact of OOD samples on DNN-based models of the visual cortex. Our analyses have two main limitations that we hope future research can address. First, we did not fine-tune the DNNs on neural data. It is possible that training these models on the specific images and/or neural data can help improve generalization. Second, we did not explore the contributions of the images being OOD for the underlying pre-trained DNNs, as we only fit the linear encoding models on train set images and neural data. Because our images were naturalistic, it is plausible that they belonged to the training distribution of the pre-trained models we used, some of which (e.g., CLIP) having hundreds of millions of images. An interesting future direction will be to examine how the model performance is affected by using out-of-distribution images for the pre-trained DNNs. These images could include those from ImageNet-P, ImageNet-C [11], and evolved images [3].

## 10   Acknowledgments

This research was partially supported by NSF grant IIS-1901030, NSF grant CCF-1231216, NIH grant R01EY026025, and NIH grant R01HD104969. We thank Pranav Misra, Fenil Doshi, Thomas Serre, and Elisa Pavarino for insightful discussions, and Harshika Bisht for design feedback on the figures. Author M.L. took the photos for the subset of images collected in the lab.

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

# Supplementary Material

## A  OOD generalization specifically challenges encoding models of ventral visual cortex.

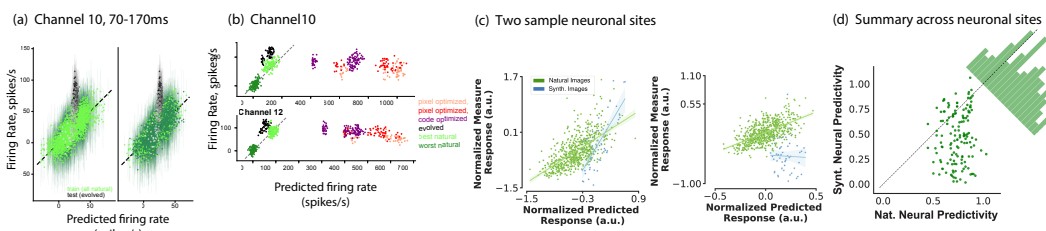

Figure S1: *OOD generalization specifically challenges encoding models of ventral visual cortex.*
(a,b) Adapted from [3]. (a) Results from an example ventral visual neuron (area PIT) show that an encoding model fit on neuronal responses to a random half of 2550 natural images accurately predicted neuronal responses to the held-out half (right subplot). However, the model consistently underpredicted responses to GAN-synthesized images evolved for the neuron [3] and did not improve from training on all the natural images (left subplot). (b) Besides underpredicting evolved-image responses, encoding models fit on natural images overpredicted neuronal responses to the models' own activation-maximization stimuli. The two subplots correspond to two example neurons. The pink to purple colors indicate how the activation-maximization stimuli were regularized (no regularization, regularization by jitter, and regularization through the latent code space of a GAN). (c,d) Adapted from [19]. (c) Results from two example ventral visual neurons (area V4) show that encoding models of neuronal responses, fit on natural images, generalized reliably to held-out, InD natural images but unreliably to OOD synthetic images. (d) For most neurons examined, encoding models performed worse on OOD than InD images.

## B  List of semantic categories in MacaqueITBench

Table S1 reports a list of all semantic categories in MacaqueITBench. The $8,233$ images correspond to 376 categories.

| Big | Big Animate | Bird | Butterfly |
|---|---|---|---|
| Cat | Dog | Face | Fish |
| Gabor | Glove | Hand | Mask |
| Misc | Non | Other | PPE |
| Print | Rodent | Starfish | Symbol |
| Toy | Turtle | abacus | accordian |
| aircompressor | airplane | ambulance | anchor |
| apple | axe | babushkadolls | babycarriage |
| babyplayard | babywalker | backgammon | backpack |
| bagel | ball | balloon | banana |
| barbiedoll | barrel | baseballbat | baseballcards |
| basket | bathsuit | battery | beaker |
| beanbagchair | bed | beermug | bell |
| bench | bike | bill | binoculars |
| birdcage | birdhouse | bones | bongo |
| bonzai | boot | boppypillow | bottle |
| bottleopener | bowl | bowlingpin | bowlofchips |
| bowtie | breadloaf | broom | bucket |
| bullet | bullhorn | button | cage |
| cake | calculator | camcorder | camera |

| | | | |
|---|---|---|---|
| candleholderwithcandle | candy | candybar | cane |
| carabiners | carfront | carseat | cashregister |
| cassettetape | ceilingfan | cellphone | chair |
| checkbook | cheese | cheesegrater | cherubstatue |
| chessboard | chocolate | christmasstocking | christmastreeornamantball |
| cigarettepack | circuitboard | clock | coatrack |
| coffeemaker | coffeemug | coffin | coin |
| collar | compass | computer | computermouse |
| cookie | cookingpan | cookpot | cooler |
| corkscrew | corset | cracker | crib |
| crossbow | crown | cupsaucer | curlingiron |
| cushion | decorativescreen | desk | doll |
| dollhouse | domino | donut | doorknob |
| doorknocker | doorwayarch | dresser | dumbbell |
| duster | dvdplayer | dynamite | earings |
| easteregg | eraser | exercise | extra |
| familiarObjects | fan | feather | filingcabinet |
| fireplace | fish hook | fishbowl | fishingpole |
| flag | flashlight | flask | fork |
| frame | fridge | frisbee | fruitparfait |
| gamehandheld | gamesboard | garbagetrash | gift |
| giftbow | glasses | globe | goggle |
| golfbag | golfball | gong | grapes |
| greenplant | grill | guitar | hairbrush |
| hairdryer | hammer | handbag | handgun |
| handheldvacuum | handkerchief | handmirror | hanger |
| hat | headband | headphone | helmet |
| highchair | hilighter | hookah | horseshoe |
| hotairballoon | hourglass | icecreamcones | iceskates |
| jack-o-lantern | jacket | juice | kayak |
| ketchupbottle | kettle | key | keyboard |
| keychain | knife | ladder | lamp |
| lantern | laptop | laudrybasket | lawnmower |
| leatherman | leaves | lei | licenseplate |
| lightbulb | lighter | lightswitch | lipstick |
| lock | log | loom | lunchbox |
| mailbox | makeupcompact | manorha | mathcompass |
| mattress | measuringtape | meat | microphone |
| microscope | microwave | motorcycle | mp3player |
| muffin | muffler | mushroom | musicstand |
| nailpolish | necklace | necktie | nest |
| nunchaku | objects | orientalplatesetting | orifan |
| pacifier | paintbrush | pants | pasta |
| patioloungechair | pda | pen | pencilsharpener |
| peppersonplate | perfumebottle | pezdispenser | phone |
| pie | pill | pillow | pipe |
| pitcher | pizza | plate | pokercard |
| powerstrip | printer | quilt | radio |
| razor | recordplayer | remotecontrol | reportfile |
| ring | ringbinder | roadsign | robot |
| rock | rollerskates | rollingpin | rosary |
| router | rug | saddle | saltpeppershake |
| sandwich | scale | scissors | scooter |
| scroll | scrunchie | seashell | seasponge |
| servingpiece | sewingmachine | shirt | shoe |
| short | shotglass | shovel | showercurtain |

| | | | |
|---|---|---|---|
| shredder | sink | sippycup | skateboard |
| slate | sleepingbag | slinky | snowglobe |
| soapdispenser | socks | sodacan | sofa |
| speakers | spicerack | spoolofstring | spoon |
| spraybottle | stamp | stapler | stool |
| stove | strainer | suit | suitcase |
| sushi | swissarmyknife | sword | tablesmall |
| tape | telescope | tennisracquet | tent |
| tire | toaster | toiletseat | tongs |
| toothbrush | toothpaste | toy | tractor |
| train | tray | tree | tricycle |
| trophy | trumpet | trunk | tupperware |
| tv | tweezer | typewriter | umbrella |
| vacuum | vase | videoGameController | wallsconce |
| washer | watch | waterbottle | watergun |
| waxseal | wheelbarrow | wheelchair | wig |
| windchime | window | wineglass | wineglassfull |
| woodboxsmall | yarn | | |

Table S1: *Images from MacaqueITBench.*

## C    Sample Images from MacaqueITBench

Fig. S2 shows sample images which were presented to Macaques to collect responses from the IT Cortex.

## D    Details on the layers used for feature extraction

For all models, we extracted features from the pre-final layer *i.e.,* the final (classification) layer was removed and features were extracted. For experiments building on features from intermediate layers, the following layers were used:

| Model | Intermediate Layer Name |
|---|---|
| resnet50_ssl | layer4.0.downsample.0 |
| resnet18_swsl | layer4.0.downsample.0 |
| resnet18 | layer4.0.downsample.0 |
| resnext101_32x16d_swsl | layer4.0.conv2 |
| noisy_student_efficient | blocks.5.2.conv_dw |
| resnext101_32x16d_swsl | layer4.0.conv2 |
| vit | blocks.10.attn.qkv |
| dinov2 | blocks.10.attn.qkv |
| bit | stages.2.blocks.13.conv1 |
| clip | transformer.resblocks.10.attn |

# E Results presented in Tabular form

| Model | Hue | Saturation | Intensity | Temperature | Contrast |
|---|---|---|---|---|---|
| resnet18 | $0.33 \pm 0.02$ | $0.36 \pm 0.02$ | $0.24 \pm 0.02$ | $0.53 \pm 0.02$ | $0.42 \pm 0.02$ |
| resnet18_swsl | $\mathbf{0.37 \pm 0.02}$ | $0.46 \pm 0.02$ | $\mathbf{0.26 \pm 0.02}$ | $\mathbf{0.67 \pm 0.03}$ | $\mathbf{0.46 \pm 0.02}$ |
| resnext101 | $0.28 \pm 0.02$ | $0.25 \pm 0.02$ | $0.17 \pm 0.01$ | $0.32 \pm 0.01$ | $0.36 \pm 0.02$ |
| noisy_student | $0.31 \pm 0.02$ | $0.21 \pm 0.01$ | $0.20 \pm 0.02$ | $0.31 \pm 0.01$ | $0.39 \pm 0.02$ |
| vit | $0.26 \pm 0.02$ | $0.20 \pm 0.01$ | $0.18 \pm 0.01$ | $0.32 \pm 0.01$ | $0.36 \pm 0.02$ |
| bit | $0.31 \pm 0.02$ | $0.33 \pm 0.02$ | $0.21 \pm 0.02$ | $0.42 \pm 0.02$ | $0.41 \pm 0.02$ |
| dinov2 | $0.28 \pm 0.02$ | $0.22 \pm 0.02$ | $0.17 \pm 0.01$ | $0.30 \pm 0.01$ | $0.34 \pm 0.02$ |
| clip | $0.34 \pm 0.02$ | $\mathbf{0.49 \pm 0.02}$ | $0.23 \pm 0.02$ | $0.66 \pm 0.03$ | $0.43 \pm 0.02$ |

Table S2: *Data from Fig.3 reported in Table form.* Neural predictivity drops significantly for all models when tested with OOD samples. The ratio of neural predictivity for OOD samples to in-distribution samples is below 1.0 for all architectures. Best performing model for each attribute is bolded.

| Model | Hue | Saturation | Intensity | Temperature | Contrast |
|---|---|---|---|---|---|
| resnet18 | $0.34 \pm 0.02$ | $0.44 \pm 0.02$ | $0.24 \pm 0.02$ | $0.52 \pm 0.03$ | $0.41 \pm 0.03$ |
| resnet18_swsl | $0.37 \pm 0.02$ | $\mathbf{0.67 \pm 0.03}$ | $\mathbf{0.27 \pm 0.02}$ | $0.75 \pm 0.03$ | $0.44 \pm 0.02$ |
| resnext101 | $0.32 \pm 0.02$ | $0.32 \pm 0.01$ | $0.21 \pm 0.02$ | $0.52 \pm 0.02$ | $0.39 \pm 0.02$ |
| noisy_student | $\mathbf{0.44 \pm 0.02}$ | $0.56 \pm 0.02$ | $0.31 \pm 0.02$ | $0.75 \pm 0.03$ | $\mathbf{0.52 \pm 0.02}$ |
| vit | $0.33 \pm 0.02$ | $0.30 \pm 0.02$ | $0.22 \pm 0.02$ | $0.48 \pm 0.02$ | $0.40 \pm 0.02$ |
| bit | $0.37 \pm 0.02$ | $0.49 \pm 0.02$ | $0.27 \pm 0.02$ | $\mathbf{0.80 \pm 0.03}$ | $0.48 \pm 0.02$ |
| dinov2 | $0.35 \pm 0.02$ | $0.41 \pm 0.02$ | $0.24 \pm 0.02$ | $0.72 \pm 0.03$ | $0.45 \pm 0.02$ |
| resnet50_ssl | $0.38 \pm 0.02$ | $0.57 \pm 0.03$ | $0.27 \pm 0.02$ | $0.73 \pm 0.03$ | $0.46 \pm 0.02$ |

Table S3: *Data from Fig.4 reported in Table form.* Neural predictivity drops on OOD samples for features extracted from different model layers as well. Best model for each attribute is bolded.

| Model | Hue | Saturation | Intensity | Temperature | Contrast |
|---|---|---|---|---|---|
| resnet18 | $0.34 \pm 0.02$ | $0.13 \pm 0.02$ | $0.64 \pm 0.03$ | $0.22 \pm 0.02$ | $0.72 \pm 0.02$ |
| resnet18_swsl | $\mathbf{0.41 \pm 0.02}$ | $\mathbf{0.15 \pm 0.01}$ | $0.71 \pm 0.03$ | $\mathbf{0.27 \pm 0.02}$ | $\mathbf{0.74 \pm 0.02}$ |
| resnext101 | $0.28 \pm 0.02$ | $0.12 \pm 0.01$ | $0.61 \pm 0.03$ | $0.20 \pm 0.01$ | $0.66 \pm 0.02$ |
| noisy_student | $0.32 \pm 0.02$ | $0.13 \pm 0.01$ | $0.46 \pm 0.02$ | $0.22 \pm 0.02$ | $0.57 \pm 0.02$ |
| vit | $0.30 \pm 0.02$ | $0.10 \pm 0.01$ | $0.53 \pm 0.02$ | $0.18 \pm 0.01$ | $0.65 \pm 0.02$ |
| bit | $0.33 \pm 0.02$ | $0.14 \pm 0.01$ | $0.61 \pm 0.02$ | $0.22 \pm 0.02$ | $0.63 \pm 0.02$ |
| dinov2 | $0.29 \pm 0.01$ | $0.11 \pm 0.01$ | $0.53 \pm 0.03$ | $0.20 \pm 0.01$ | $0.63 \pm 0.02$ |
| clip | $0.38 \pm 0.02$ | $0.14 \pm 0.01$ | $\mathbf{0.77 \pm 0.04}$ | $0.23 \pm 0.02$ | $0.73 \pm 0.02$ |

Table S4: *Data from Fig.5 reported in Table form.* Neural predictivity drops on OOD samples for the *high* hold-out strategy as well. Best performing model for each attribute is bolded.

# F Additional results with hold-out strategies

In the main paper, we presented results with two hold out strategies—high and low. Here, we present additional results with held out strategies. Firstly, we confirmed that specialized domain generalization architectures also struggle with the low-hold out strategy as shown in Fig. S3. Furthermore, we present results with the third hold-out strategy outlined in the paper. We refer to this as the Mid hold out strategy as samples between the 42.5 and the 67.5 percentile of every OOD attribute are held out as the test set. As shown in Fig. S4, across all architectures and OOD attributes, models suffer to generalize to OOD samples for the Mid hold out strategy.

# G Additional results with intermediate layers

In the main paper we presented results for models trained with intermediate layers for the high hold out strategy. Here we provide additional results with models that use intermediate layers of DNNs as

feature extractors. In Fig. S5 and Fig. S6 we report results for the *low* and *mid* hold-out strategies respectively.

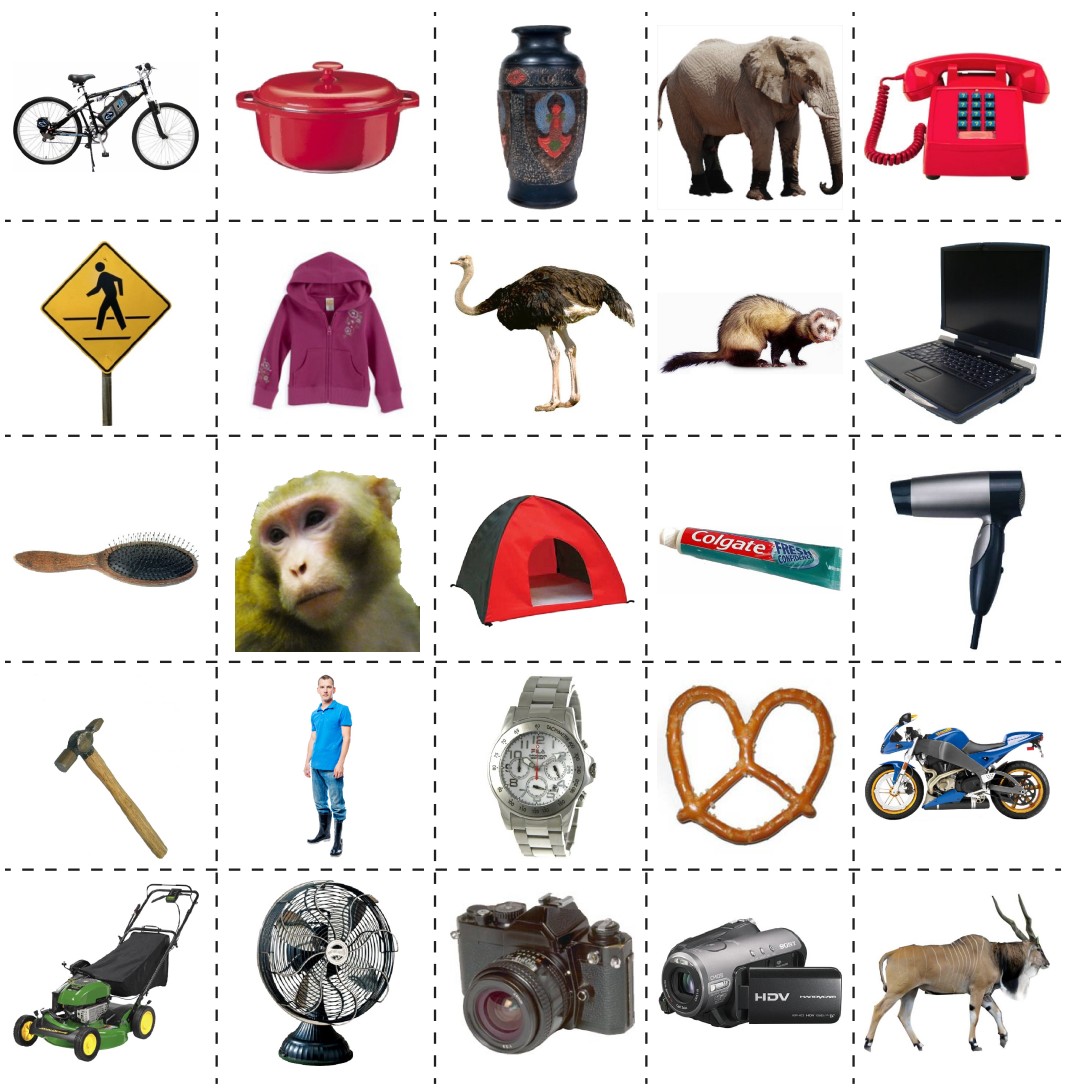

Figure S2: *Example images from MacaqueITBench.*

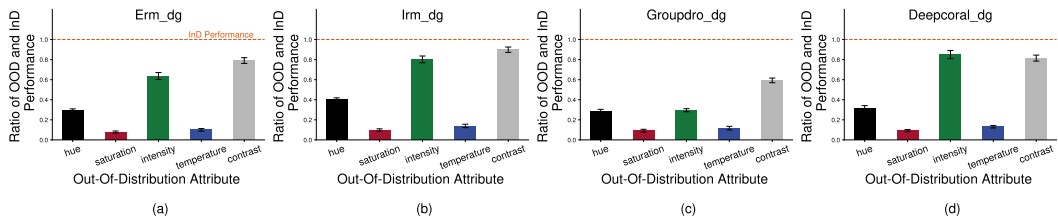

Figure S3: *Neural predictivity drops for specialized domain generalization approaches with the low hold-out strategy as well.* Neural predictivity is reported on OOD test splits constructed using the *low* hold-out strategy. Ratio of OOD and in-distribution neural predictivity is below $1.0$ for all approaches and all image-computable attributes, panels (a-d). Thus, these approaches do not generalize well to OOD splits constructed with the *low* hold-out strategy as well.

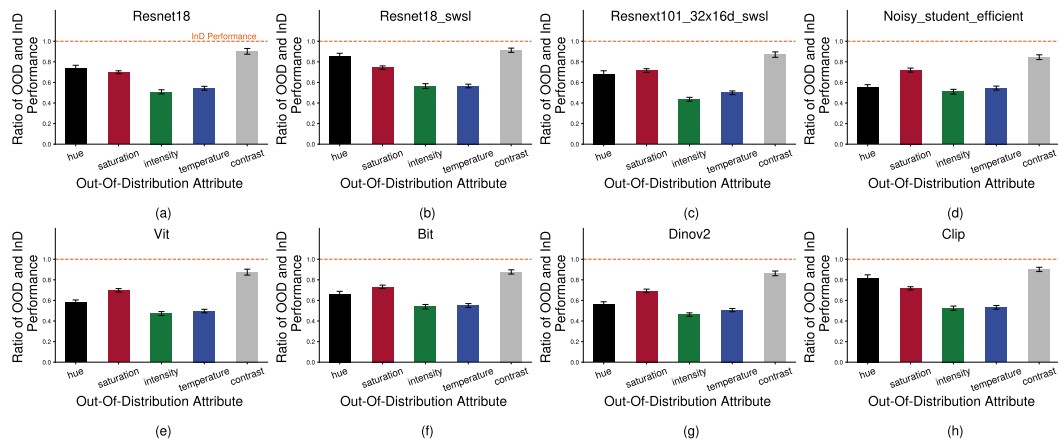

Figure S4: *Neural predictivity drops for Mid hold-out strategy as well.* For all architectures, across multiple OOD shifts, performance on OOD is worse than in-distribution samples for the Mid hold-out strategy as well.

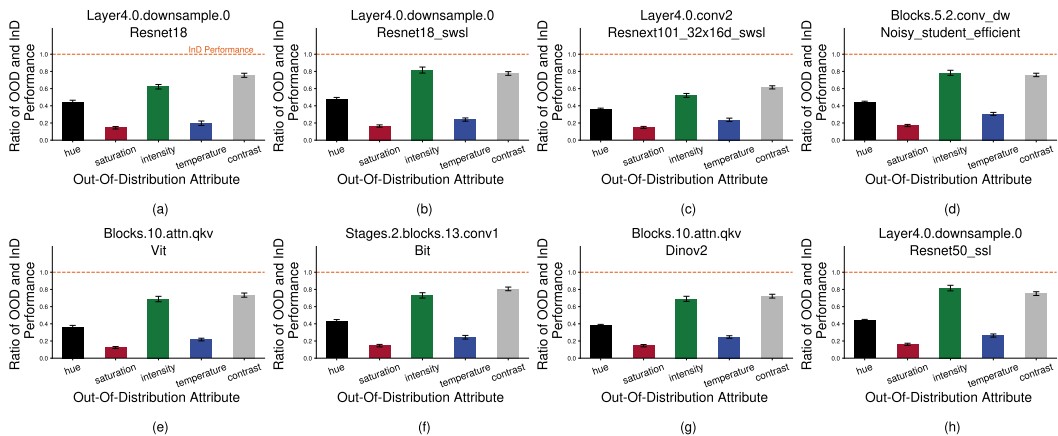

Figure S5: *Neural predictivity drops for low hold-out strategy for intermediate layer features as well.* For all architectures, across multiple OOD shifts, performance on OOD is worse than in-distribution samples for the low hold-out strategy for image features extracted from intermediate DNN layers as well.

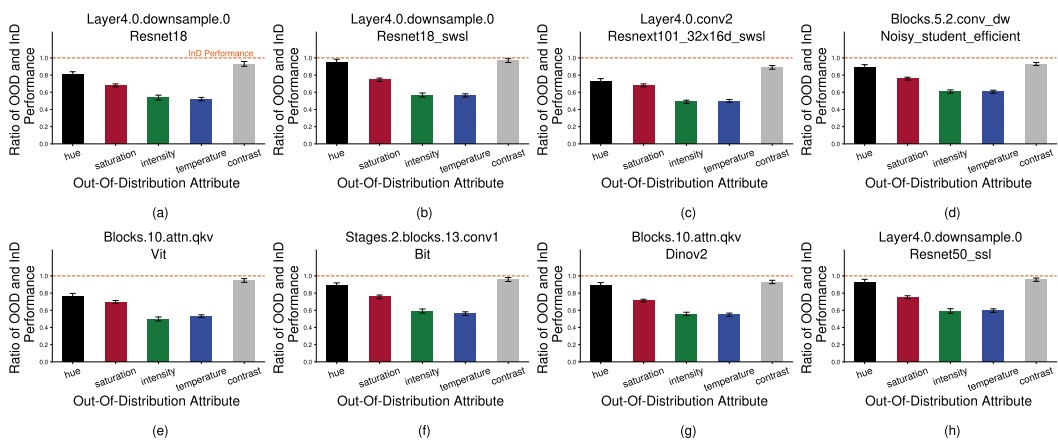

Figure S6: *Neural predictivity drops for mid hold-out strategy for intermediate layer features as well.* For all architectures, across multiple OOD shifts, performance on OOD is worse than in-distribution samples for the mid hold-out strategy for image features extracted from intermediate DNN layers as well.

