# OpenReview forum: "Benchmarking Out-of-Distribution Generalization Capabilities of DNN-based Encoding Models for the Ventral Visual Cortex."
_NeurIPS.cc/2024/Datasets_and_Benchmarks_Track — NeurIPS 2024 Track Datasets and Benchmarks Poster_

### Official Review · Reviewer_daFw · 2024-07-25
**Review of Benchmarking Out-of-Distribution Generalization Capabilities of DNN-based Encoding Models for the Ventral Visual Cortex.**

**Rating:** 8
**Confidence:** 4
**Correctness:** Yes, the methods and approach are okay.
**Clarity:** Yes

**Review:**

Deep learning based models are increasingly used to understand the behavior of the biological networks. It is now a common practice to fit the neural recording using DNN based representation and use the resultant models to predict responses to novel stimuli. In this context, robustness of the models is very important as the responses of such networks are used to test hypothesis about the behavior of the biological neural networks. This study establishes a very important limitation of such practices. Using feature based splits authors have successfully demonstrated the fragile nature of the models are their failure to generalize to shifts in low level image features. This dataset and current performance results are very helpful in advancing the modeling approaches and making them more robust.

**Strengths:**

1. A large scale database of neural recordings useful for investigating deep learning approaches for model fitting.
2. Demonstration of the fragility and lack of generalization of the popular deep learning based encoders for predicting neural responses

**Additional Feedback:**

I would like thank the authors for this paper.

**Documentation:**

Yes

**Limitations:**

There are no major limitations, it would be nice to have further analysis on why the models are not generalizing well.

**Opportunities For Improvement:**

It would be great if authors could investigate the potential reasons for this lack of OOD generation.

**Relation To Prior Work:**

Authors have discussed the relevant literature appropriately.

**Summary And Contributions:**

Authors have developed a large-scale dataset of neural population responses from the macaque inferior temporal (IT) cortex as natural images were presented to the monkeys. By carefully splitting the images into different sets based on the image features such as hue and contrast, authors have demonstrated that deep learning models used to predict neural responses fail to generalize well and are sensitive to changes in the features. This is an important and counter intuitive results as we would expect deep neural networks to be robust and invariant to such shifts.

---

> ### Author Rebuttal · Authors · 2024-08-16
>
> # Thank you for your review
>
> We are grateful for your constructive review. We agree that ‘why models struggle to generalize OOD’ is an important question concerning deep learning models, in particular models of the ventral visual cortex. We take first steps toward answering this question by contributing a large-scale dataset, a benchmarking framework for OOD neural predictivity, and an initial explanation.
>
> Specifically, our results suggest that the distance between the training and testing distributions, i.e. the amount of distribution shift, is a key factor governing the degree of generalizaiton, and we demonstrate an image-computatble method to  quantitatively predict the degree of model generalization [Fig. 6, Section 7.2]. We believe these results will inform future work toward a deeper understanding that synthesizes ideas from both OOD generalization and model-brain representation similarity.

---

> > ### Comment · Reviewer_daFw · 2024-08-27
> > **Thank you**
> >
> > I would like to thank the authors for their rebuttal, my score remains the same. I believe this work in this direction is important and will look forward to deeper analysis of the failures as well as more formal and in depth treatment of the in distribution/out of distribution constructs going beyond the features such as contrast.

---

### Official Review · Reviewer_VCLi · 2024-07-25
**An interesting study of OOD generalization of DNNs in predicting neuronal responses from the visual cortex.**

**Rating:** 6
**Confidence:** 4
**Correctness:** No
**Clarity:** Yes

**Review:**

Please find my detailed comments in the sections below.

**Strengths:**

(+) It's new to study the OOD generalization capabilities of DNNs in neural predictivity.

(+) This work presents clear and illustrative instructions on how to curate and split the data.

(+) The benchmarking includes most of the prevalent DNN methods.

(+) The analysis is also comprehensive and interesting.

**Additional Feedback:**

N/A

**Documentation:**

Yes

**Limitations:**

Yes

**Opportunities For Improvement:**

1. The benchmarking does not consider OOD generalization methods. It's suggested to refer to [1,2] to choose proper OOD generalization baselines to examine whether the considered problem could already be resolved by OOD generalization methods.

2. The OOD splits are not natural. When applying DNNs to the task, there will not be artificial OOD splits in terms of image properties. It is unclear to what extent the split scheme in this benchmark could reflect the challenges of this task.

**References**

[1] In Search of Lost Domain Generalization, ICLR'20

[2] WILDS: A Benchmark of in-the-Wild Distribution Shifts, ICML'21.

**Relation To Prior Work:**

Yes

**Summary And Contributions:**

This work studies the OOD generalization capabilities of DNNs in predicting neuronal responses from the visual cortex. The authors construct a benchmark called MacaqueITBench, a large-scale dataset of neural population responses from the macaque inferior temporal (IT) cortex to over 300, 000 images, comprising 8, 233 unique natural images presented to seven monkeys over 109 sessions. They use MacaqueITBench to split the samples into ID and OOD subsets, with respect to image attributes such as Contrast, Saturation and etc.. They evaluate several DNNs such as ResNet18 and find that the DNNs can only retain as little as 20% performance for OOD splits, compared to ID splits.

---

> ### Author Rebuttal · Authors · 2024-08-16
>
> # Additional results added with the OOD Generalization methods specified by the reviewer.
>
> We present these additional experiments below:
>
> 1. **“benchmarking does not consider OOD generalization methods”**
>
> Our focus here is on computational models of biological vision (e.g., [1,2,3] ). OOD generalization methods cited in the review [4,5] were not designed or tested on neural predictivity.
>
> However, we appreciate the reviewer’s suggestion and have now **added results with these specialized domain generalization approaches. Please find them in the attached PDF.** We will include these results in the camera-ready version. Our findings hold true for these specialized approaches as well—neural predictivity is substantially lower on OOD test samples despite the use of specialized approaches designed for OOD generalization.
>
> In Neuroscience, there has been enthusiasm about the claim that deep neural networks (DNNs) are adequate and may even be sufficient models of the visual cortex [1,2,3]. This work demonstrates the pitfalls in this idea by showing that simple distribution shifts break these models of the brain. The fact that such models display poor generalization in OOD shows that they are insufficient models of the brain.
>
> 2. **“OOD splits are not natural”**
>
> Our main contribution here is not to provide a new methodology to improve OOD performance in machine learning. Instead, we take inspiration from machine learning to emphasize critical implications for the understanding of brain function. The OOD splits we tested represent cornerstone features studied in Vision Science and Computer Science [6,7,8,9,10]. The ability to bridge these OOD splits is critical for any model that claims to explain ventral cortex function. For example, a model of the visual cortex should be able to predict neuronal responses at noon and also at 8PM, when the contrast and color changes drastically. We test both OOD features here.
>
> We do not claim that the splits we tested are exhaustive. For example, we now add two additional splits. First, we held out a random subset of categories as a test set and trained on the remaining categories. Second, we held out Food-related categories as a test set and trained on non-Food categories. These are relevant because models frequently need to generalize to novel categories. For both these OOD splits, all models failed to generalize—ratio of OOD and In-Distribution neural predictivity was well below 1.0, in line with our results for other OOD splits.
>
> | Model            | Random Categories | Food Categories |
> |------------------|-------------------|-----------------|
> | CLIP             | 0.80 ± 0.01       | 0.23 ± 0.02     |
> | DinoV2           | 0.77 ± 0.02       | 0.19 ± 0.01     |
> | Noisy Student    | 0.78 ± 0.02       | 0.20 ± 0.02
> | ResNet18         | **0.83 ± 0.01**       | **0.23 ± 0.02**    |
> | ResNet50_SSL     | 0.80 ± 0.01       | 0.20 ± 0.02     |
> | ResNext101       | 0.71 ± 0.01       | 0.16 ± 0.01     |
>
> Finally, we publicly share the data to facilitate future work to evaluate other shifts and develop better models of the ventral visual cortex. Developing better models of the ventral visual cortex could also lead to better OOD generalization in machine learning.
>
> References:
>
> 1. Schrimpf, Martin, et al. "Brain-score: Which artificial neural network for object recognition is most brain-like?." BioRxiv(2018): 407007.
> 2. Schrimpf, Martin, et al. "Integrative benchmarking to advance neurally mechanistic models of human intelligence." Neuron108.3 (2020): 413-423.
> 3. Willeke, Konstantin F., et al. "The sensorium competition on predicting large-scale mouse primary visual cortex activity." arXiv preprint arXiv:2206.08666 (2022).
> 4. Gulrajani, Ishaan, and David Lopez-Paz. "In search of lost domain generalization." arXiv preprint arXiv:2007.01434(2020).
> 5. Koh, Pang Wei, et al. "Wilds: A benchmark of in-the-wild distribution shifts." International conference on machine learning. PMLR, 2021.
> 6. Nauhaus, Ian, et al. "Stimulus contrast modulates functional connectivity in visual cortex." Nature neuroscience 12.1 (2009): 70-76.
> 7. Louie, Kenway, and Paul W. Glimcher. "Normalization principles in computational neuroscience." Oxford research encyclopedia of neuroscience. 2019.
> 8. Kreiman, Gabriel. Biological and computer vision. Cambridge University Press, 2021.
> 9. Livingstone, Margaret, and David Hubel. "Segregation of form, color, movement, and depth: anatomy, physiology, and perception." Science 240.4853 (1988): 740-749.
> 10. MS, LIVINGSTONE. "Anatomy and physiology of a color system in the primate visual cortex." J. Neurosci. 4 (1984): 309-356.

---

> > ### Comment · Reviewer_VCLi · 2024-08-21
> >
> > Thank you for the detailed rebuttal and additional experiments. I am increasing my score by assuming that the discussions above and the experiments will be added to the paper. Moreover, it would be appreciated if there were more explanations on the motivation of the OOD splits so that it would be more impactful and easy-to-follow even for people with limited background knowledge in this field.

---

> > > ### Author Rebuttal · Authors · 2024-08-21
> > >
> > > Thank you for the feedback. We will expand the explanation and citations to motivate the different OOD splits. We will also ask colleagues with different expertise to comment on the manuscript to make sure that the motivation is clear for people with limited background in the field. Again, we appreciate the feedback.

---

### Official Review · Reviewer_hU8b · 2024-08-02

**Rating:** 6
**Confidence:** 3
**Correctness:** The claims made in the submission are…
**Clarity:** The presentation is clear and easy to…

**Review:**

Pros:
1. The proposed benchmark MacaqueITBench is designed for the important problem of OOD in ventral visual cortex.
2. The introduction of the Macaque-ITBench dataset provides a substantial resource for studying neural responses to a wide variety of images, covering multiple visual cortex areas.
3. The paper systematically examines the impact of various types of distribution shifts (contrast, hue, intensity, temperature, and saturation) on model performance.
4. The dataset and benchmarks are publicly available, promoting transparency and further research.
5. The presentation is clear and easy to follow.

Cons:
1. The discussion and conclusions are not as insightful as expected. For example, stronger distribution shifts cause severe performance drop is common knowledge in OOD related works. Thus more specific and insightful discussions would improve the contribution of the paper.
2. The experiments in the main text and appendix are displayed through figures. While figures can often reflect simple patterns and conclusions, the lack of tables with specific values results in insufficient precision in the paper. Including tables with specific values would more accurately present the experimental results. Additionally, some numbers in the figures are too small to be easily read.

**Strengths:**

1. The Macaque-ITBench dataset offers a substantial resource for studying neural responses to a wide variety of images across multiple visual cortex areas.
2. The paper systematically investigates the impact of various distribution shifts (contrast, hue, intensity, temperature, and saturation) on model performance.
3. The presentation is clear and easy to understand.

**Additional Feedback:**

None

**Documentation:**

Yes.

**Ethics:**

None.

**Limitations:**

The authors have adequately addressed the limitations and potential negative societal impact of their work.

**Opportunities For Improvement:**

1. If more analysis on OOD issues related to the Ventral Visual Cortex could be conducted, and insights directly related to the Ventral Visual Cortex could be provided, rather than insights similar to most existing conclusions, it would significantly enhance the paper's contribution.
2. If some experimental results could be presented in tables, or if specific experimental results could be shown in tables in the appendix, it would help readers understand the experimental results more precisely.

**Relation To Prior Work:**

Yes.

**Summary And Contributions:**

The paper investigates how well deep neural network (DNN) models generalize when predicting neuronal responses from the macaque visual cortex under various distribution shifts. The authors introduce Macaque-ITBench, a large-scale dataset of neural responses to over 300,000 images from the macaque inferior temporal (IT) cortex. They find that DNN models, while effective on in-distribution data, perform significantly worse on out-of-distribution (OOD) data, with performance drops as high as 80%. The paper also shows that the cosine distance between image representations extracted from a pre-trained object recognition model is a strong predictor of neural predictivity under different distribution shifts.

---

> ### Author Rebuttal · Authors · 2024-08-16
>
> # The tables and results requested under Opportunities for Improvement have been added.
>
>
> We have added the additional results and tables requested in your review:
>
> 1. **“More specific and insightful discussions [...]”** and **“insights directly related to the ventral visual cortex.”**
>
> We agree that the detrimental effects of strong distribution shifts are well-known in OOD-related machine learning literature. However,  OOD generalization has seldom been studied in the context of modeling biological neural responses despite the large volume of work (e.g., [1,2,3,4]; reviewed in [5]). The ability of models to generalize (OOD) is especially relevant to the ventral visual cortex due to acute limitations on the amount of neural data feasible to collect (not much more than 1 to 10k unique images, including repeat presentations needed to combat neural stochasticity).
>
> In this data-limited regime, most images of interest will necessarily remain out-of-domain *even if* we had foreknowledge of the test distribution (e.g.,10k unique images mean only 10 images per category for the 1,000 ImageNet categories, insufficient to cover the distribution). Indeed, the poor OOD generalization of current neural encoding models is limiting their scientific utility. A practical case is the need to accurately predict maximally activating images for neurons, **a case we now summarize in the new Fig. 13 (see attached PDF)**. Thus, our contribution is to take inspiration from the machine learning literature on OOD generalization and show the relevance to computational models of neuroscience, an active area of research.
>
> We also compared models on an established, in-distribution benchmark (BrainScore) and on our OOD benchmark. These results, presented below,  further support our finding that current DNNs are insufficient models of the Ventral Visual Cortex—models that perform better on the in-distribution BrainScore benchmark did not perform better on OOD shifts (all Spearman rank correlations p > 0.05).
>
> | Model      | BrainScore | OOD Hue | OOD Saturation | OOD Intensity | OOD Temp. | OOD Contrast | OOD Average |
> |------------|------------|---------|----------------|---------------|-----------|--------------|-------------|
> | BiT        | 0.33       | 0.31    | 0.33           | 0.21          | 0.42      | 0.41         | 0.33        |
> | ResNet18   | 0.35       | 0.33    | 0.36           | **0.24**          | 0.53      | 0.42         | 0.37        |
> | CLIP       | 0.47       | **0.34**    | **0.49**           | 0.23          | **0.66**      | **0.43**         | **0.43**        |
> | ResNext101 | 0.49       | 0.28    | 0.25           | 0.17          | 0.32      | 0.36         | 0.32        |
> | ViT        | **0.51**       | 0.26    | 0.20           | 0.18          | 0.32      | 0.36         | 0.30        |
>
>
>
> These contributions are in addition to our two other contributions–1) publishing the largest publicly available dataset to help the community advance this research; 2) proposing a quantitative way to predict OOD performance drop.
>
> 2. **“If experimental results could be presented in tables..“**
>
> As requested, we now present our results also as tables **(see attached PDF)**. We will include these tables in the supplement for the camera-ready version.
>
> **References:**
> 1. Yamins, Daniel LK, et al. "Performance-optimized hierarchical models predict neural responses in higher visual cortex." Proceedings of the national academy of sciences 111.23 (2014): 8619-8624.
> 2. Schrimpf, Martin, et al. "Brain-score: Which artificial neural network for object recognition is most brain-like?." BioRxiv(2018): 407007.
> 3. Schrimpf, Martin, et al. "Integrative benchmarking to advance neurally mechanistic models of human intelligence." Neuron108.3 (2020): 413-423.
> 4. Willeke, Konstantin F., et al. "The sensorium competition on predicting large-scale mouse primary visual cortex activity." arXiv preprint arXiv:2206.08666 (2022).
> 5. Serre, Thomas. "Deep learning: the good, the bad, and the ugly." Annual review of vision science 5.1 (2019): 399-426.

---

> > ### Author Response · Authors · 2024-08-26
> > **Gentle reminder to review the rebuttal before the Aug 31st deadline.**
> >
> > Dear Reviewer, with the deadline of the discussion phase (August 31st) fast approaching, we wanted to kindly remind you to review the additional experiments and results we’ve incorporated based on your suggestions. Your feedback would be greatly appreciated.

---

### Author Rebuttal · Authors · 2024-08-16

#  Added four new experiments, tables, and discussion.

We thank the reviewers for their questions and suggestions. To address them, we added new results and discussion, as summarized below. Detailed responses are provided in our individual reviewer responses and attached PDFs, and will also be included in the camera-ready version:

**Reviewer hU8b:**

1. We added two sets of results offering additional insights into the Ventral Visual Cortex.
2. We now present experimental results also in table form as requested by the reviewer.

**Reviewer VCLi:**

1. We added results with specialized Domain Generalization approaches as the reviewer requested. Our findings extend to these specialized approaches.
2. We added results with two more ‘natural’ OOD splits.

**Reviewer daFw:**

1.As the reviewer requested, we discuss potential mechanisms driving poor OOD performance.

---

### Decision · Program_Chairs · 2024-09-26

**Decision:**

Accept (Poster)

**Comment:**

The paper introduces a new dataset for out-of-distribution (OOD) generalization in the task of neural response prediction from the visual cortex. It collects 8,233 unique natural images through an experiment involving seven monkeys. Distribution shifts are created by splitting the data based on attributes such as image contrast, hue, intensity, etc. The paper concludes that stronger distribution shifts lead to poorer generalization performance for DNN-based encoding models, which are evaluated in this study. The authors have well addressed the concerns raised by reviewers, and I would vote for accepting this paper.

However, I would strongly recommend the authors to address the following concerns in their final version. While the paper highlights an important issue in neural response prediction affected by distribution shift, it remains unclear from the rebuttal how this task is distinct within the broader OOD literature, particularly in the field of computer vision. Given that OOD generalization is already well-studied in machine learning, with mature benchmarks like WILDS, it is crucial to clarify the unique challenges of neural response prediction, which are insufficiently addressed in the paper. Additionally, a systematic review and evaluation of existing OOD approaches is lacking, though some methods were supplemented during the rebuttal phase. And the distribution shifts are synthetically produced by unnatural split of data.